# Multilesion Segmentations in Patients with Intracerebral Hemorrhage: Reliability of ICH, IVH and PHE Masks

**Estelle Vogt** [1],*, **Ly Huong Vu** [1], **Haoyin Cao** [1], **Anna Speth** [2], **Dmitriy Desser** [2], **Frieder Schlunk** [2,3], **Andrea Dell'Orco** [2] and **Jawed Nawabi** [1,3]

1   Department of Radiology, Charité School of Medicine and University Hospital Berlin, 10117 Berlin, Germany
2   Department of Neuroradiology, Charité School of Medicine and University Hospital Berlin, 10117 Berlin, Germany
3   Berlin Institute of Health (BIH), BIH Biomedical Innovation Academy, 10178 Berlin, Germany
*   Correspondence: estelle.vogt@charite.de

**Abstract:** Background and Purpose: Fully automated methods for segmentation and volume quantification of intraparenchymal hemorrhage (ICH), intraventricular hemorrhage extension (IVH), and perihematomal edema (PHE) are gaining increasing interest. Yet, reliabilities demonstrate considerable variances amongst each other. Our aim was therefore to evaluate both the intra- and interrater reliability of ICH, IVH and PHE on ground-truth segmentation masks. Methods: Patients with primary spontaneous ICH were retrospectively included from a German tertiary stroke center (Charité Berlin; January 2016–June 2020). Baseline and follow-up non-contrast Computed Tomography (NCCT) scans were analyzed for ICH, IVH, and PHE volume quantification by two radiology residents. Raters were blinded to all demographic and outcome data. Inter- and intrarater agreements were determined by calculating the Intraclass Correlation Coefficient (ICC) for a randomly selected set of patients with ICH, IVH, and PHE. Results: 100 out of 670 patients were included in the analysis. Interrater agreements ranged from an ICC of 0.998 for ICH (95% CI [0.993; 0.997]), to an ICC of 0.979 for IVH (95% CI [0.984; 0.993]), and an ICC of 0.886 for PHE (95% CI [0.760; 0.938]), all *p*-values < 0.001. Intrarater agreements ranged from an ICC of 0.997 for ICH (95% CI [0.996; 0.998]), to an ICC of 0.995 for IVH (95% CI [0.992; 0.996]), and an ICC of 0.980 for PHE (95% CI [0.971; 0.987]), all *p*-values < 0.001. Conclusion Manual segmentations of ICH, IVH, and PHE demonstrate good-to-excellent inter- and intrarater reliabilities, with the highest agreement for ICH and IVH and lowest for PHE. Therefore, the degree of variances reported in fully automated quantification methods might be related amongst others to variances in ground-truth masks.

**Keywords:** computed tomography; intracranial hemorrhage; intraventricular hemorrhage; perihematomal edema; interrater reliability; intrarater reliability; ground-truth; deep learning

## 1. Introduction

Intracerebral hemorrhage (ICH) is the most severe form of stroke with a one-month morbidity and mortality approaching 50% and in the course of time exceeding 75% [1–3]. Non-contrast CT (NCCT) is widely used in the clinical diagnosis of ICH because of its high imaging speed and high sensitivity and specificity in the detection of stroke [4]. Many increasingly proposed quantitative-imaging markers and artificial intelligence (AI) methods based on NCCT are used for the prediction of hematoma expansion (HE), prognosis of ICH, the extent of secondary intraventricular hemorrhage (IVH), and the evaluation of perihematomal edema (PHE) [4]. Especially fully automated deep learning methods are in high demand as they may offer a time efficient and accurate volume analysis of ICH, IVH, and PHE. Accurate and reliable quantification of these volumes will be paramount to their utility as measures of treatment effect in future clinical studies [5]. Yet, the accuracy has shown to vary among the three proposed lesion classes with restricted comparability due to different quantification methods [6–8]. Our objective was to evaluate differences in the

level of agreement between manually derived multiclass segmentation masks of ICH, PHE, and IVH. Therefore, we hypothesized differences amongst ground-truth segmentations masks for ICH, IVH, and PHE. To test and evaluate this hypothesis, the interrater and intrarater reliability was assessed for all three classes of ICH, IVH, and PHE to allow a better interpretation of automated segmentation tools as described in the above.

## 2. Methods

### 2.1. Study Population

We retrospectively studied the databases at Charité University Hospital Berlin, a German tertiary center, between January 2016 and June 2020. Inclusion criteria were defined as (1) primary, spontaneous, non-traumatic ICH, (2) age > 18, (3) baseline non-contrast computed tomography (NCCT) images acquired within 24 h from onset/last seen well (LSW). Primary spontaneous ICH were included despite severity and size, and anticoagulant treatment. Patients with secondary causes of ICH due to head trauma, brain tumor, vascular malformation, primary intraventricular hemorrhage, or secondary ICH from hemorrhagic transformation of ischemic infarction were excluded. Additional clinical variables were extracted from patients' clinical records (Table 1). A final subset of 100 patients was randomly selected by calculating a formula with the INDEX function (Figure 1) [7–10]. This single center retrospective study was approved by the ethics committee (Ethik-Kommission der Charité Berlin; running number EA1/035/20) and written informed consent was waived by the institutional review boards. All study protocols and procedures were conducted in accordance with the Declaration of Helsinki. Due to the retrospective nature of the study, patient consent was not needed.

**Table 1.** Demographic characteristics.

| Baseline Clinical and Imaging Characteristics | All = 100 |
|---|---|
| Clinical characteristics | |
| Age median, (IQR) | 75.5 (65–78) |
| Male, *n* (%) | 83 (83%) |
| Hypertension, *n* (%) | 68 (68%) |
| Diabetes mellitus, *n* (%) | 12 (12%) |
| Initial GCS, median (IQR) | 11 (10–14) |
| RRsys, median (IQR) | 155.5 (0–189.25) |
| Anticoagulation, *n* (%) | 34 (34%) |
| Antiplatelet Treatment, *n* (%) | 16 (16%) |
| Imaging characteristics | |
| Bleeding location, *n* (%)<br>lobar<br>basal ganglia<br>thalamus<br>brainstem/Pons<br>cerebellar | <br>92 (92%)<br>8 (8%)<br>0<br>0<br>0 |
| Black hole sign, *n* (%) | 13 (13%) |
| Blend sign, *n* (%) | 6 (6%) |
| Hypodensities, *n* (%) | 13 (13%) |
| Island sign, *n* (%) | 12 (12%) |
| Spot sign, *n* (%) | 8 (8%) |

**Table 1.** *Cont.*

| Baseline Clinical and Imaging Characteristics | All = 100 |
|---|---|
| Surgical Treatment, *n* (%) | |
| Supratentoriell craniectomy | 26 (26%) |
| Infratentoriell craniectomy | 3 (3%) |
| EDV | 13 (13%) |
| Minimally invasive surgery | 4 (4%) |
| Clinical Outcome | |
| mRS > 3, *n* (%) | 45 (45%) |
| mRS < 3, *n* (%) | 55 (55%) |

Legend: indicates percentage; IQR indicates interquartile range; n indicates absolute number. GCS, Glasgow Come Scale; IQR, interquartile range; mRS, modified Rankin Scale; RRsys, systolic arterial blood pressure; EDV, external ventricular drain.

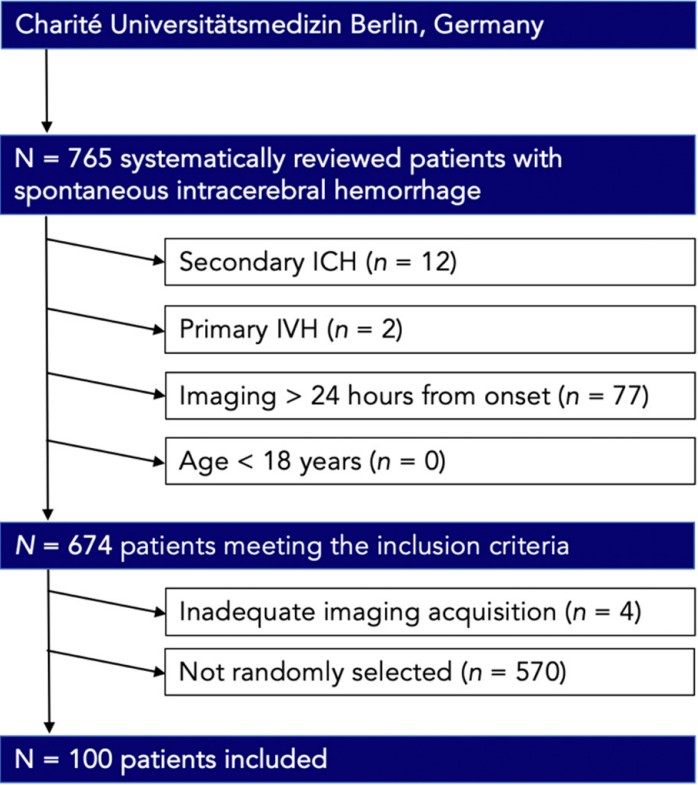

**Figure 1.** Patient flowchart indicative of the final subset according to the inclusion criteria. IVH, intraventricular hemorrhage.

*2.2. Image Acquisition*

CT scans were performed on a 80 slice scanner (Toshiba Aquilion Prime, Tochigi, Japan) with the following imaging parameters: NCCT with 120 kV, 300 mA, 5.0 mm slice reconstruction; CTA: 100–120 kV, dosis-modulated between 260–300 mA, 1.0 mm slice reconstruction, 5 mm MIP reconstruction with 1 mm increment, 0.5 mm collimation, 0.64 pitch, separate reconstruction kernels (brain, FC21; bone, FC30) at the same thickness (1 and 5 mm gapless), 60 mL highly iodinated contrast medium, and 30 mL NaCl flush at 4 mL/s; the scan started 6 s after bolus tracking at the level of the ascending aorta. CTA was committed to further analysis of this study.

## 2.3. Image Analysis

Imaging data were retrieved in the Digital Imaging and Communications in Medicine (DICOM) format from the local picture archiving and communication system (PACS) servers, anonymized in compliance with the local guidelines, and transformed into Neuroimaging Informatics Technology Initiative (NifTI) format. One radiology fellow (J.N., 5 years of experience in stroke imaging) assessed and documented (1) the presence of intraventricular hemorrhage and (2) ICH location on admission and follow-up NCCT scans. Supratentorial bleedings in cortical and subcortical location were classified as lobar whether hemorrhages involving the thalamus, basal ganglia, internal capsule and deep periventricular white matter were classified as deep [11]. Infratentorial bleedings were classified within the brainstem and pons or cerebellum [12]. In the following two radiology residents (E.V. and L.V., both 2 years of experience in stroke imaging) segmented ICH, IVH and PHE (manual planimetric measurement) on the basis of the original NCCT images [13,14]. Regions of interest (ROIs) were delineated using ITK-SNAP 3.8.0 Software [14–16]. The ROI histogram for ICH and IVH segmentation was sampled between 20 and 80 Hounsfield units (HU) to exclude voxels that likely belong to cerebrospinal fluid or calcification. The ROI histogram for PHE segmentation was sampled between 0 and 30 HU to exclude voxels that likely belong to leukoaraiosis [13,17]. NCCT markers were rated according to the proposed international consensus definition of Morotti et al. [17,18]. Both raters (E.V. and L.V.) independently reviewed images in a random order, blind to all demographic and outcome data and were not involved in the clinical care of assessment of the enrolled patients. Images were randomized and presented again to the second rater (E.V.) one month later for a second reading, to minimize recall of the patients' follow-up scans.

## 2.4. Statistical Analysis

Data were tested for normality and homogeneity of variance using histogram plots and Shapiro–Wilk test. Descriptive statistics are presented as counts (percentages [%]) for categorical variables, mean (standard deviation [SD]) for continuous normally distributed variables, and medians (interquartile range [IQR]) for non-normal continuous variables. Friedman's test was used to test for pairwise comparisons of ICH, PHE, and IVH volumes [19,20]. Interrater and intrarater agreement was calculated and expressed as Intraclass Correlation Coefficient (ICC) with stratified kappa with 95% upper and lower confidence intervals (CI) [21,22]. For intrarater agreement, ICC calculations were stratified by across two readings from one rater (E.V.), whereas for interrater agreement, ICC calculations were stratified by each combination of reading (reading 1 or reading 2) from pairs of the two raters (E.V. and L.V.). Analyses were performed using the statistical software package SPSS version 25® (IBM Corporation, Armonk, NY, USA). A statistically significant difference was accepted at a *p*-value less than 0.05. Calculation of the formula with the INDEX function was conducted in Microsoft Exel® (Microsoft Corporation (2018). Microsoft Excel, Armonk, NY, USA).

## 3. Results

Our analysis included NCCT images of 100 patients with acute primary ICH and secondary IVH who fulfilled the inclusion criteria. Table 1 shows the baseline clinical features of the patient collective. The median age was 75.5 years (IQR 65–78) with 17 (17%) female patients. Patients presented upon admission with a median GCS of 11 (IQR 10–14) and median systolic blood pressure of 155.5 (IQR 0–189.25). Of the total 100, 68 (68%) patients were diagnosed with hypertension and 12 (12%) with diabetes mellitus. Some of the patients were pre-treated with oral blood thinners, 16 (16%) took antiplatelets and 34 (34%) anticoagulation. Most bleedings (92%) were in the lobe. The most frequent observed NCCT markers were the black hole sign and the hypodensity sign, in 13 (13%) patients each. Poor clinical outcome (mRS > 3 in 90 days) was observed in 45% of the patients.

The median volumes for ICH were for rater one 17.325 mL (IQR 7.57–40.38) in the first reading and 18.33 mL (IQR 7.665–41.808) in the second reading and the second rater

measured 19.825 mL (IQR 8.17–42.84; Table 2). The corresponding values for PHE were 12.6 mL (IQR 5.12–23.39), 11.81 mL (5.42–24.64), and 16.55 (IQR 7.84–28.76). The median volumes for IVH of the two intrarater segmentations were 6.34 mL (IQR 2.33–13.15) and 6.5 mL (IQR 1.96–12.02), and 6.11 mL for the second rater (IQR 2.26–12.31). The Friedman test revealed significant differences in volumes among ICH, PHE, and IVH in all three ratings (Table 2).

**Table 2.** Comparison of measured volumes.

| Radiological Features | | | |
| --- | --- | --- | --- |
| | **Rating 1 (Rater 1, First Rating)** | **Rating 2 (Rater 1, Second Rating)** | **Rating 3 (Rater 2)** | ***p*-Value \*** |
| ICH volume [mL], median (IQR) | 17.325 (7.57–40.38) | 18.33 (7.665–41.808) | 19.825 (8.17–42.84) | <0.001 |
| PHE volume [mL], median (IQR) | 12.6 (5.12–23.39) | 11.81 (5.42–24.64) | 16.55 (7.84–28.76) | <0.001 |
| IVH volume [mL], median (IQR) | 6.34 (2.33–13.15) | 6.5 (1.96–12.02) | 6.11 (2.26–12.31) | 0.005 |

Legend: ICH, intracerebral hemorrhage; IVH, intraventricular hemorrhage, PHE, perihematomal edema, SD, standard deviation. \* *p*-value given for statistical analysis of Friedman test.

Across all three ratings, both intra- and interclass agreement showed good to excellent results which are given in detail in Tables 3 and 4. Segmentations for ICH had the highest agreement with ICC values ranging from 0.997 for the intrarater reliability and 0.998 for the interrater reliability, all *p*-values < 0.001. Segmentations for IVH showed an ICC of 0.995 for intrarater reliability and an ICC of 0.979 for interrater reliability, all *p*-values < 0.001. The lowest correlation was analyzed for PHE with ICC values of 0.98 for intraclass reliability and 0.886 for interclass reliability, all *p*-values < 0.001. Illustrative examples for high and low intra- and interrater agreements for ICH, PHE, and IVH segmentation are given in Figures 2 and 3.

**Table 3.** Intraclass Correlation of one rater for semimanual volume segmentation of intracerebral hemorrhage and intraventricular hemorrhage stratified across two readings.

| Intraclass Correlation | | | | |
| --- | --- | --- | --- | --- |
| **Region** | **ICC \*** | **95% Lower CI** | **95% Upper CI** | ***p*-Value** |
| ICH (*n* = 100) | 0.997 | 0.996 | 0.998 | <0.001 |
| PHE (*n* = 100) | 0.980 | 0.971 | 0.987 | <0.001 |
| IVH (*n* = 100) | 0.995 | 0.992 | 0.996 | <0.001 |

Legend: Intraclass agreement of semimanual volume quantification of intracerebral hemorrhage and intraventricular hemorrhage on computed tomography specified with intraclass correlation coefficient (ICC) with 95% confidence interval (CI). \* Stratified ICC across one rater in two ratings.

**Table 4.** Interclass Correlation of one rater for semimanual volume segmentation of intracerebral hemorrhage and intraventricular hemorrhage stratified across two readings.

| Interclass Correlation | | | | |
| --- | --- | --- | --- | --- |
| **Region** | **ICC \*** | **95% Lower CI** | **95% Upper CI** | ***p*-Value** |
| ICH (*n* = 100) | 0.998 | 0.993 | 0.997 | <0.001 |
| PHE (*n* = 100) | 0.886 | 0.760 | 0.938 | <0.001 |
| IVH (*n* = 100) | 0.979 | 0.984 | 0.993 | <0.001 |

Legend: Interclass agreement of semimanual volume quantification of intracerebral hemorrhage and intraventricular hemorrhage on computed tomography specified with intraclass correlation coefficient (ICC) with 95% confidence interval (CI). \* Stratified ICC across two raters.

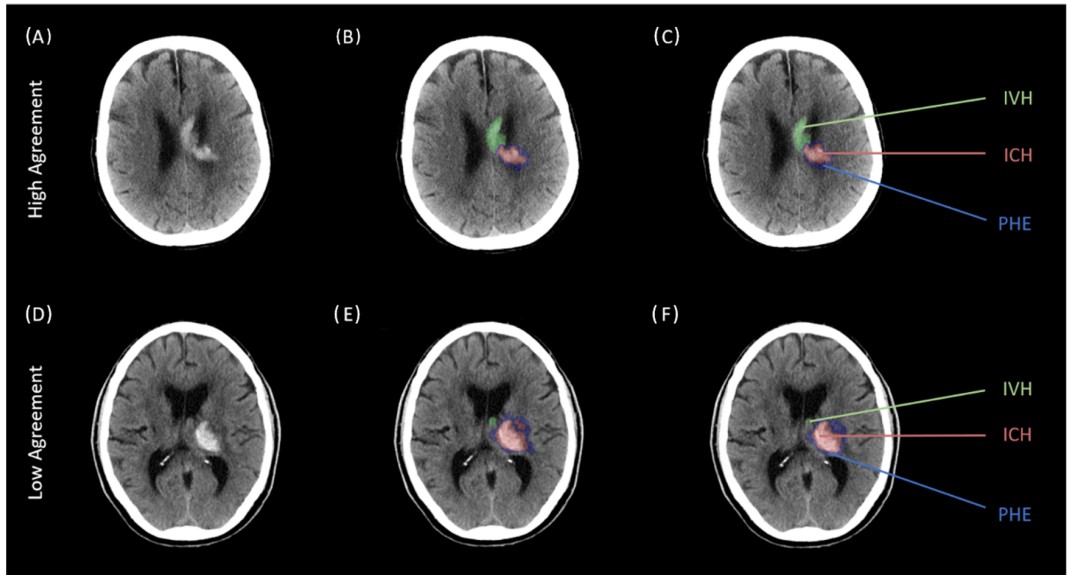

**Figure 2.** Two representative cases of high and low intrarater agreement comparing two different segmentations of one rater across two separate readings. Legend: Two representative cases of intrarater agreement for the segmentation of intracerebral hemorrhage (ICH; red), perihematomal edema (PHE; blue), and intraventricular hemorrhage (IVH; green) across two separate readings of one rater. (**A–C**): Above case representing high agreements with corresponding NCCT scan (**A**), first reading (**B**) and second reading (**C**). (**D–F**): Below case representing low agreements with corresponding NCCT scan (**D**), first reading (**E**) and second reading (**F**). ICH indicates intracerebral hemorrhage; IVH, intraventricular hemorrhage, PHE, perihematomal edema.

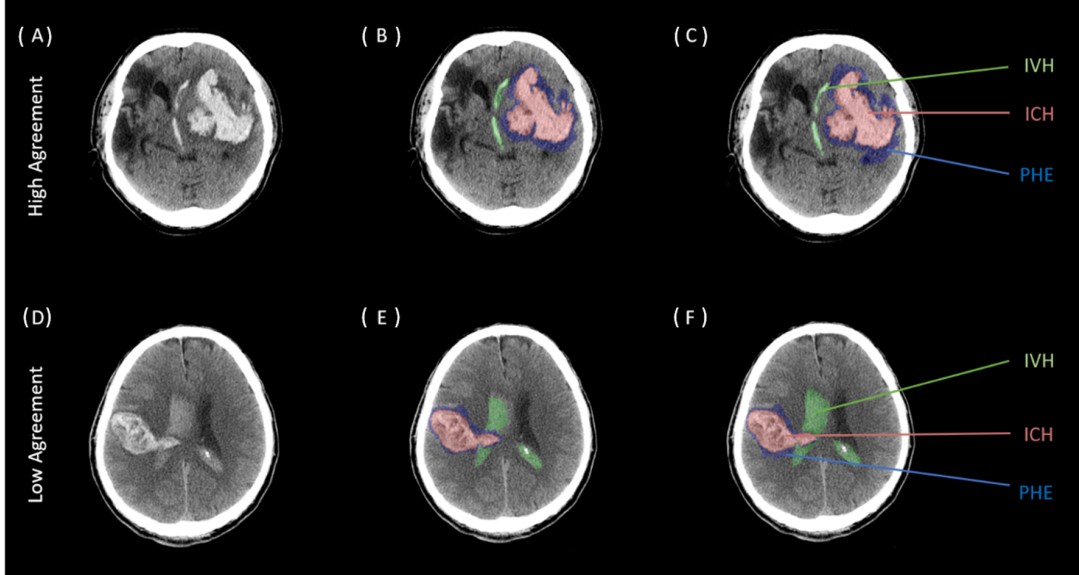

**Figure 3.** Two representative cases of high and low interrater agreement comparing two different segmentations of two raters across two separate readings. Two representative cases of interrater agreement for the segmentation of intracerebral hemorrhage (ICH; red), perihematomal edema (PHE; blue), and intraventricular hemorrhage (IVH; green) across two separate readings of two rater. (**A–C**): Above case representing high agreements with corresponding NCCT scan (**A**), first reading (**B**) and second reading (**C**). (**D–F**): Below case representing low agreements with corresponding NCCT scan (**D**), first reading (**E**) and second reading (**F**). ICH indicates intracerebral hemorrhage; IVH, intraventricular hemorrhage; PHE, perihematomal edema.

## 4. Discussion

Our study was designed and powered to assess measurement equivalency between the ICH, IVH, and PHE volumes of manual quantification methods. Results of our study are unique in terms of the assessment of both inter- and intrarater agreements for all three classes. Our findings revealed overall good-to-excellent correlation for the segmentations of ICH, IVH and PHE. When assessed with intrarater ICC, correlations were almost equally excellent for all three classes with an ICC > 0.9. In comparison, interrater correlations for PHE segmentations were relatively lower when compared to ICH and IVH, although volume analysis was statistically different among all three classes. Our study agrees with the results that the segmentation of PHE is challenging due to its unclear boundary to the surrounding white matter [23]. Especially alterations within the white matter due to microangiopathic changes impede segmentation accuracy as density changes may even decrease upon visual inspection. To overcome this setback-effect, increasing clinical studies have utilized the method proposed by Volbers et al., allowing an improved differentiation between microangiopathic changes and PHE by using a window setting of 5–33 Hounsfield Unit (HU)(13). Evaluation of IVH was comparably higher than that of PHE and yet did not exceed that of ICH. This finding is most probably related to the difficult visual differentiation of IVH with the blending of cerebrospinal fluid—especially when the hematoma appears at a hyperacute state. Statistical evaluation of the performance of deep learning-based segmentation methods is often presented with the dice correlation coefficient (DICE) which is based on the spatial overlap between two sets of segmentations [24]. Additional calculation of the concordance correlation coefficient (CCC) may be adopted to measure the agreement between the predicted lesion volume and the (semi-)manually derived ground-truth lesion volume. Therefore, the performance is limited by the quality of the ground truth masks on which they are trained [5]. To our knowledge, Zhao et al. were the first to present a fully automated segmentation method including all three lesions' classes of ICH, IVH, and PHE [7]. Reported CCC of ICH and IVH (CCCs $\geq$ 0.98) were excellent and also that of PHE (CCCs $\geq$ 0.92) demonstrated good concordance. The CCC is a simple index of how well a new measurement reproduces a gold standard measurement with established guidelines for how to interpret the magnitude of concordance, but with no direct link between the value of the index and the magnitude of inter-quantification method measurement discrepancy [25]. Manually generated ground truth masks by Zhao et al. [7] were evaluated in 20 subjects for intrarater and 40 subjects for interrater reliability—ranging from median Dice scores of 0.85 for ICH, 0.73 for IVH, and 0.69 for PHE. Regarding interrater reliability, the median Dice scores were 0.87, 0.73, and 0.68 for ICH, IVH, and PHE, respectively. In conclusion, the level of agreement between ground truth masks gradually decreased from ICH to IVH to PHE. For this reason, it is important to consider inter- and intraquantification discrepancies to allow for a better interpretation of potential bias of AI methods' performance as they may produce results that differ from the true underlying estimate [25]. Other future directions that are related to the improvement of AI methods for ICH segmentations may also explore the effect of negative training subjects on the AI's performance [26–28]. Several limitations deserve to be addressed. One limitation of this study is the retrospective single-center design of our study. Secondly, the proposed selection method in our study did not ascertain for the risk of selection bias which is further limited by the number of subjects included—admittedly at the expense of the generalization of the results as infratentorial bleedings were unfortunately not represented and may have contributed to different reliability agreements. Future research using a larger sample size is recommended in order to investigate reliabilities in infratentorial ICH. Therefore, a study should ideally consider the total number of patients included. Finally, we did not present results according to the level of expertise of the raters.

In conclusion overall reliability of segmentations for ICH, IVH and PHE were good-to-excellent yielding lowest accuracies for PHE. To our best of knowledge, this is the largest study evaluating accuracies of multilesion segmentations in patients with acute ICH [7,8]. Our results permit legitimate discussion of inference about inter- and intraquantification

method measurements when interpreting AI-methods utilizing multilesion segmentations in ICH patients.

**Author Contributions:** Conceptualization, J.N. and E.V.; methodology, J.N. and E.V.; software, J.N. and E.V.; validation, J.N. and E.V.; formal analysis, J.N. and E.V.; investigation, J.N. and E.V.; resources, J.N. and E.V.; data curation, J.N. and E.V.; writing—original draft preparation, E.V.; writing—review and editing, L.H.V., H.C., A.S., D.D., F.S. and A.D.; visualization, J.N.; supervision, J.N.; project administration, J.N.; funding acquisition, J.N. All authors have read and agreed to the published version of the manuscript.

**Funding:** No funding available.

**Institutional Review Board Statement:** This multicenter retrospective study was approved by the ethics committee (Charité Berlin, Germany [protocol number EA1/035/20], University Medical-Center Hamburg, Germany [protocol number WF-054/19] University Hospital Muenster, Germany [protocol number 2017-233-f-S], and IRCCS Mondino Foundation, Pavia, Italy [protocol number 20190099462] and written informed consent was waived by the institutional review boards. All study protocols and procedures were conducted in accordance with the Declaration of Helsinki.

**Informed Consent Statement:** Patient consent was not needed due to the retrospective nature of the study.

**Data Availability Statement:** The datasets that support the findings of our study are available upon reasonable request from the corresponding author, however prior approval of proposals apply by our institution's data security management and a signed data sharing agreement will then be approved.

**Acknowledgments:** Jawed Nawabi is grateful for being part of the BIH Charité—Digital Clinician Scientist Program funded by Charité—Universitaetsmedizin Berlin, the Berlin Institute of Health and the German Research Foundation (DFG, Deutsche Forschungsgemeinschaft). Frieder Schlunk is grateful for being part of the BIH Charité –Clinician Scientist Program funded by Charité—Universitaetsmedizin Berlin, the Berlin Institute of Health.

**Conflicts of Interest:** All other authors have nothing to disclose.

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
