# Peer review of "Multilesion Segmentations in Patients with Intracerebral Hemorrhage: Reliability of ICH, IVH and PHE Masks"

_tomography, doi:10.3390/tomography9010008_

Round 1

Reviewer 1 Report

Paper is well written, however there are some parts that need a revision.

- Lines 43-45: "Accurate, reliable, and efficient quantification of these volumes will be paramount to their utility as measures of treatment effect in future clinical studies"  What do authors mean with this sentence?

- Lines 48-53: "We aimed to investigate whether the reported variances may be linked... interpretation of automated segmentation tools as described in the above." What is the primary and secondary objective? revise.

- "To our knowledge, Zhao et al. were the first to present a fully automated segmentation method including all three lesions' classes of ICH, IVH, and PHE. " Add reference and discuss more about this.

- Lines 229-231: Textiloma mimicking intracranial bleeding can be a condition that should be included and discussed. These papers should be considered: -- Postoperative Textiloma Mimicking Intracranial Rebleeding in a Patient with Spontaneous Hemorrhage: Case Report and Review of the Literature. Case Rep Neurol. 2020 Jan 9;12(1):7-12. doi: 10.1159/000505233  -- Textiloma (gossypiboma) mimicking recurrent intracranial abscess. BMC Res Notes. 2015 Aug 30;8:39.

- Add a conclusion section. What is new about this article compared to the previous ones?

- Table 2 shows that ICH, PHE and IVH volume affect p value. Discuss more these results.

Author Response

Response to Editors and Reviewers

Reviewer #1:

Paper is well written, however there are some parts that need a revision.

Re: We thank the Reviewer for his or her interest in our work and for the helpful comments that will improve the manuscript. We have tried to do our best to respond to the points raised and we appreciate the opportunity to clarify our research objectives and results.

As indicated below, we have checked all the general and specific comments provided by the Referees and have made necessary changes accordingly to their indications.

- Lines 43-45: "Accurate, reliable, and efficient quantification of these volumes will be paramount to their utility as measures of treatment effect in future clinical studies"  What do authors mean with this sentence?

Re: We thank you the reviewer for this question. Accurate refers to the diagnostic accuracy of such algorithms evaluated by accuracy or receiver operating curve characatersittics by calculating the area under the curve. Reliable refers to the stability and generalization of the approach e.g. to an independent data set. Efficient refers to the advantage of a timely improved quantification which we agree might not necessarily be part of an implementation criterion for clinical trials. As such, we have adjusted the sentence to the following: “Accurate and reliable quantification (…)”

- Lines 48-53: "We aimed to investigate whether the reported variances may be linked... interpretation of automated segmentation tools as described in the above." What is the primary and secondary objective? revise.

Re: We thank you the reviewer for the opportunity to specify this important matter. We have revised the sentence into the following: “Our objective was to evaluate differences in the level of agreement between manually derived multiclass segmentation masks of ICH, PHE, and IVH.”

As our second objective more a discussion of our results - to whether our results may be linked to the variances reported in automatically derived multiclass lesions masks of recent AI quantification methods – we omitted this part from this part.

- "To our knowledge, Zhao et al. were the first to present a fully automated segmentation method including all three lesions' classes of ICH, IVH, and PHE. " Add reference and discuss more about this.

Re: We apologize for the missing reference which we have added. In fact, we have discussed the paper in the followings lines as follows:

“To our knowledge, Zhao et al. were the first to present a fully automated segmentation method including all three lesions' classes of ICH, IVH, and PHE6. Reported CCC of ICH and IVH (CCCs ≥ 0.98) were excellent and also that of PHE (CCCs ≥ 0.92) demonstrated good concordance. The CCC is a simple index of how well a new measurement reproduces a gold standard measurement with established guidelines for how to interpret the magnitude of concordance, but with no direct link between the value of the index and the magnitude of inter-quantification method measurement discrepancy24. Manually generated ground truth masks by Zhao et al6. were evaluated in 20 subjects for intrarater and 40 subjects for interrater reliability – ranging from median Dice scores of 0.85 for ICH, 0.73 for IVH, and 0.69 for PHE. Regarding inter-rater reliability, the median Dice scores were 0.87, 0.73, and 0.68 for ICH, IVH, and PHE, respectively. In conclusion, the level of agreement between ground truth masks gradually decreased from ICH to IVH to PHE. For this reason, it is important to consider inter- and intraquantification discrepancies to allow for a better interpretation of potential bias of AI methods’ performance as they may produce results that differ from the true underlying estimate24.“

- Lines 229-231: Textiloma mimicking intracranial bleeding can be a condition that should be included and discussed. These papers should be considered: -- Postoperative Textiloma Mimicking Intracranial Rebleeding in a Patient with Spontaneous Hemorrhage: Case Report and Review of the Literature. Case Rep Neurol. 2020 Jan 9;12(1):7-12. doi: 10.1159/000505233  -- Textiloma (gossypiboma) mimicking recurrent intracranial abscess. BMC Res Notes. 2015 Aug 30;8:39.

Re: We thank you the author for these two interesting articles which we have integrated in the discussion in line 345 ff:

Other future directions that are related to the improvement of AI methods for ICH segmentations may also explore the effect of negative training subjects on the AI’s performance25,26,27.“

- Add a conclusion section. What is new about this article compared to the previous ones?

Re: Thank you for this comment. A conclusion had already been included in the manuscript (line 424 ff). Regarding the novelty of our results, we have added a paragraph which reads now as follows:

“In conclusion overall reliability of segmentations for ICH, IVH and PHE were good-to-excellent yielding lowest accuracies for PHE. To our best of knowledge, this is the largest study evaluating accuracies of multilesion segmentations in patients with acute ICH.  Our results permit legitimate discussion of inference about inter- and intraquan-tification method measurements when interpreting AI-methods utilizing multilesion segmentations in ICH patients.”

- Table 2 shows that ICH, PHE and IVH volume affect p value. Discuss more these results.

Re: We thank you the reviwer for pointing out this important result which have added as follows to the discussion paragraph (line 267 ff)

“In comparison, interrater correlations for PHE segmentations were relatively lower when compared to ICH and IVH, although volume analysis was statistically different among all three classes.“

Reviewer 2 Report

Short paper, that is carefully and clearly written.  A few questions remain despite the comprehensive discussion:

1. use leukoaraiosis instead of leucariosis

2. The patient's selection process ist not clear. Why were the 100 patients selected out from initial 670 and how was this type of bias handled? It would add to to better understanding when a more detailed description of patients included in the study  vs. the excluded is given.

Author Response

Response to Editors and Reviewers

Reviewer #2:

Short paper, that is carefully and clearly written.  A few questions remain despite the comprehensive discussion:

Re: We thank the Reviewer for his or her interest in our work and for the helpful comments that will improve the manuscript. We have tried to do our best to respond to the points raised and we appreciate the opportunity to clarify our research objectives and results.

As indicated below, we have checked all the general and specific comments provided by the Referees and have made necessary changes accordingly to their indications.

  1. use leukoaraiosis instead of leucariosis

Re: We thank you the reviewer for pointing out this spelling error which we have revised accordingly (line 104).

  1. The patient's selection process ist not clear. Why were the 100 patients selected out from initial 670 and how was this type of bias handled? It would add to to better understanding when a more detailed description of patients included in the study vs. the excluded is given.

Re: We thank you the reviewer for this very important and helpful comment.

Referring to the number of patients selected: We selected 100 out of 670 subjects as this number represents a reasonable number to conduct a stable deep learning analysis according to our own experience. Unfortunately, no current definition is given as to how many subjects should be selected in order to evaluate deep learning approaches. The same limitation refers to the evaluation of interrater reliabilities of ground truth masks. The authors Zhao et al. (https://doi.org/10.1007/s00330-020-07558-2) for example, selected 40 out of 380 subjects (ratio 1:10) for their reliability analysis. In this aspect, our study adds important findings as our selected subjects were 5 times higher within a total study cohort double as high compared to the one described in the study of Zhao et al. (ratio 1:7). In line with this, our ratio is similar to the one suggested by Dhar et al. who proposed an algorithm for intracerebral hemorrhage (ICH) and perihematomal edema (PHE) segmentation in which the authors analyzed 20 out of 124 subjects for their reliability analysis (ratio 1:6) (https://doi.org/10.1161/STROKEAHA.119.027657). We have added this discussion to the discussion paragraph (line 252 ff):

“To our best of knowledge, this is the largest study evaluating accuracies of multilesion segmentations in patients with acute ICH. “

Referring to the selection method: A randomly selected and prespecified number of subjects was defined to evaluate interrater reliabilities; a method described in many similar studies and exemplary shown in the following clinical studies:

1) In the study of Wolff et al. analyzing the collateral status: “Baseline CTA scans with an intracranial anterior occlusion from the MR CLEAN study (n=500) were used. For each core lab CS, ten CTA scans with sufficient quality were randomly selected.” https://doi.org/10.1007/s00234-022-02984-z

2) In the study of Zhao et al analyzing ICH lesions masks: “To test the intra-rater reliability, repeated delineation of 20 randomly chosen cases was performed 4 weeks apart by one rater (X.Z.). For assessing the interrater reliability, two raters independently performed segmentations in 40 randomly selected cases.” https://doi.org/10.1007/s00330-020-07558-2

3) In the study of Zhao et al analyzing ICH lesions masks: “In 20 cases, we had one rater repeat delineation 1 week apart to assess intrarater reliability, while in 40 cases we had 2 raters independently perform manual segmentation of hemorrhage and PHE.” https://doi.org/10.1161/STROKEAHA.119.027657

In addition to the selection method, a random selection of the 100 subjects was conducted by calculating a formula with the INDEX function in Microsoft Exel. This INDEX function allows to get a random value from a list.  https://support.microsoft.com/en-us/office/index-function-a5dcf0dd-996d-40a4-a822-b56b061328bd

We have added this to the methods section as follows (line 66 ff): “A final subset of 100 patients was randomly selected by calculating a formula with the INDEX function.” And line 158 ff: “Calculation of the formula with the INDEX function was conducted in Microsoft Exel® (Microsoft Corporation (2018). Microsoft Excel).” The above mentioned references have also been added.

We absolutely agree with the reviewer that our method does not ascertain for the risk of selection bias and that it is further limited by the number of subjects included - admittedly at the expense of the generalization of our results as infratentorial bleedings were unfortunately not represented. We have added these points to as limitations to the discussion (line 248 ff) as the following:

“ The proposed selection method in our study did not ascertain for the risk of selection bias which is further limited by the number of subjects included - admittedly at the expense of the generalization of the results as infratentorial bleedings were unfortunately not represented and may have contributed to different reliability agreements. Future research using a larger sample size is recommended in order to investigate reliabilities in infratentorial ICH. Therefore, a study should ideally consider the total number of patients included.”

Finally, a patients flowchart diagram has been added to the revised manuscript to support the above mentioned explanations.

                Figure 1: Patient flowchart indicative of the final subset according to the inclusion criteria. IVH, intraventricular hemorrhage.

The topic of noncontrast computed tomography (NCCT) markers in patients with intracerebral hemorrhage is relevant and should be of interest to the readers. The paper is generally well writen and includes a large and relevant patient material.

Re: We thank you the reviewer very much for this overall positive feedback and hope that we were able to address all concerns properly. We especially thank the reviewer for allowing us to revise our method section to increase the reader’s understanding of our study. 

Round 2

Reviewer 1 Report

It is good.